# Micro-Machining Characteristics in High-Speed Magnetic Abrasive Finishing for Fine Ceramic Bar

**Joonhyuk Song [1], Takeo Shinmura [2], Sang Don Mun [3,4,\*] and Minyoung Sun [3,4,\*]**

[1] Korea Institute of Carbon Convergence Technology, 10, Unam-ro, Deokjin-gu, Jeonju 54853, Korea; songjh@kctech.re.kr

[2] Department of Mechanical Systems Engineering, Utsunomiya University, 7-1-2, Yoto, Utsunomiya, Tochigi 321-8585, Japan; shinmura@cc.utsunomiya-u.ac.jp

[3] Division of Mechanical Design Engineering, Jeonbuk National University, 664-14, Duckjin-dong, Jeonju 561-756, Korea

[4] Jeonbuk National University, International Offshore Wind Institute, 664-14, Duckjin-dong, Jeonju 561-756, Korea

\* Correspondence: msd@jbnu.ac.kr (S.D.M.); smy5439@jbnu.ac.kr (M.S.); Tel.: +82-63-270-4762 (S.D.M.); +82-63-270-2453 (M.S.)

**Abstract:** The research aims to describe the micro-machining characteristics in a high-speed magnetic abrasive finishing, which is applicable for achieving the high surface accuracy and dimensional accuracy of fine ceramic bars that are typically characterized by strong hardness and brittle susceptibility. In this paper, the high-speed magnetic abrasive finishing was applied to investigate how the finishing parameters would have effects on such output parameters as surface roughness, variation of diameters, roundness, and removed weight. The results showed that, under variants of diamond abrasives sizing between (1, 3 and 9 μm), 1 μm showed comparatively good values as for surface roughness and roundness within shortest processing time. When the optimal condition was used, the surface roughness $R$a and roundness (LSC) were improved to 0.01 μm and 0.14 μm, respectively. The tendency of diameter change could be categorized into two regions—stable and unstable. The finding from the study was that the performance of ultra-precision processing linear controlling was possibly achievable for the stable region of diameter change, while linearly controlling diameters in the workpiece.

**Keywords:** micro-machining; high-speed magnetic abrasive finishing; fine ceramic bar; surface roughness ($R$a); roundness (LSC); removed weight (RW); diameter (d)

## 1. Introduction

Amidst the remarkable advancement in industrial technologies, intensive attempts have been made to achieve better mechanical properties and performances, even under harsher conditions, especially by implementing fine ceramics to light-weighted, minimalized, and ultra-precision parts applied to the fields of information, telecommunications, nano-bio industry, photoelectric, and aeronautical/aerospace industries [1,2]. To keep pace with the trend, the machining field adopts more advanced methods for processing materials, such as grinding, polishing, lapping, superfinishing, honing, burnishing, etc. [3–7].

However, when the workpiece is a difficult-to-machine material, such as fine ceramic bar with 3 mm in diameter, these methods become unachievable, because they used high pressures during processing, which damage the surface to be processed [8]. Furthermore, the micro-cracks can be formed on the surface of workpiece after processing by the conventional methods [9,10]. Fine ceramic materials are of outstanding difficulty in application by conventional grinding or polishing methods

due to their physical properties of high brittleness and hardness though admitting that grinding for improving surface flatness is normally applied. Singh et al. [9] concluded that high surface accuracy or dimensional accuracy of difficult-to-machine material (e.g., ceramic, silicon nitride, and silicon carbide) are difficult to achieve by these conventional methods. New different advanced processing methods have been adopted for replacing these conventional methods to overcome these problems.

Unlike the conventional methods, the magnetic abrasive finishing process is one of the advanced finishing methods, which is widely put into practice as an ultra-precision technology by benefiting the magnetic field while using magnetic abrasive materials so as to generate abrading forces that are applicable to the surfaces of parts or objects [11,12].

In this regard, the objectives of this study lie in providing the characteristics of ultra-precision finishing under variables of surface roughness, tendency of diameter change, roundness, and removed weight behavior by actually incorporating magnetic abrasive finishing with high speed and tightly modulated conditions against hard and brittle fine ceramic bars.

## 2. Principle of Micro-Machining Process for Fine Ceramic Bar

Figure 1 shows the schematic diagram of the external magnetic abrasive finishing process, where mixed type magnetic abrasive tools (electrolytic iron powder and diamond paste) are filled between the S and N poles to form a flexible magnetic abrasive brush (FMAB) to the direction of magnetic density flux. The targeted round workpiece is inserted inside a flexible magnetic abrasive brush and then rotated with a high rotational speed, while vibration to the axial direction is applied onto the workpiece. The finishing force affecting the target object is exerted to its surface along with the direction of magnetic force line by magnetic particles, accordingly causing abrasion to be under process.

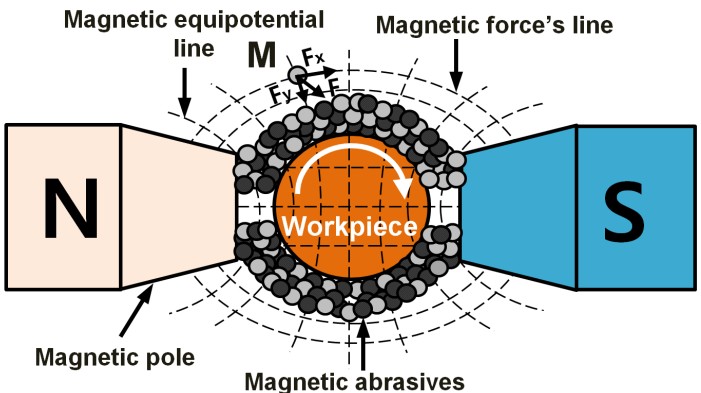

**Figure 1.** Schematic diagram of external micro-machining process using magnetic field.

## 3. Experimental Configuration and Method

Figure 2 shows the schematic view of the apparatus for magnetic abrasive finishing. A fine ceramic round bar is inserted inside the gap of magnetic poles and then rotated with a high rotational speed up to 30,000 rpm at the same time together with 12 Hz of vibration being applied to finish the bar. A pneumatic spindle was used to generate the high rotational speed of workpiece at 30,000 rpm, which a spindle controller can adjust. Two sets of Nd-Fe-B permanent magnets with an electric slider being equipped for making it vibrate. This magnetic section is composed of magnetic poles, Nd-Fe- B permanent magnet, and York (See Figure 2). The mixed type of magnetic abrasive tools was filled in between magnetic poles-made of SS400 steel for better processing, and then a fine ceramic round bar was inserted before applying rotational and vibrating movement, as shown in Figure 1. The mixed type abrasive that was adopted for the experiment was gained by mechanically mixing diamond paste and electrolytic iron particles including 0.2 mL of grinding fluid as lubricant. The directions of a magnetic pole approaching toward the target workpiece for processing, where the A direction, not C, was made for best ultra-precision effectivity, while the magnetic poles were moved to the B direction

after processing (see A-view of Figure 2). Herein, a pneumatic cylinder was utilized for effectuating moving forces. The key factors that were measured after processing included roughness around the micro-machined surface, such as its roundness, resultant change in micro-diameter, and minute finishing volume. Table 1 shows the material characteristics in fine ceramics applied. Table 2 describes the detailed experimental conditions set for this study. In this study, eight numbers of fine ceramic bars were prepared as the workpieces (dimension: $\phi 3 \times 60$ mm), and their accuracy characteristics before and after processing, such as surface roughness, variation of diameters, roundness, and removed weight, were measured by surface roughness tester (Mitutoyo SJ-400, Mitutoyo, Sakado, Japan), laser diameter tester (Mitutoyo LSM-501S, Mitutoyo, Sakoda, Japan), roundness tester (Mitutoyo RA-114, Mitutoyo, America corparation, Sakado, Japan), and weight tester (Shimadzu AUW220D, Shimadzu, Kyoto, Japan), respectively. The workpieces were measured three times at different positions at every 30 s of processing time in order to determine the average value of these accuracies.

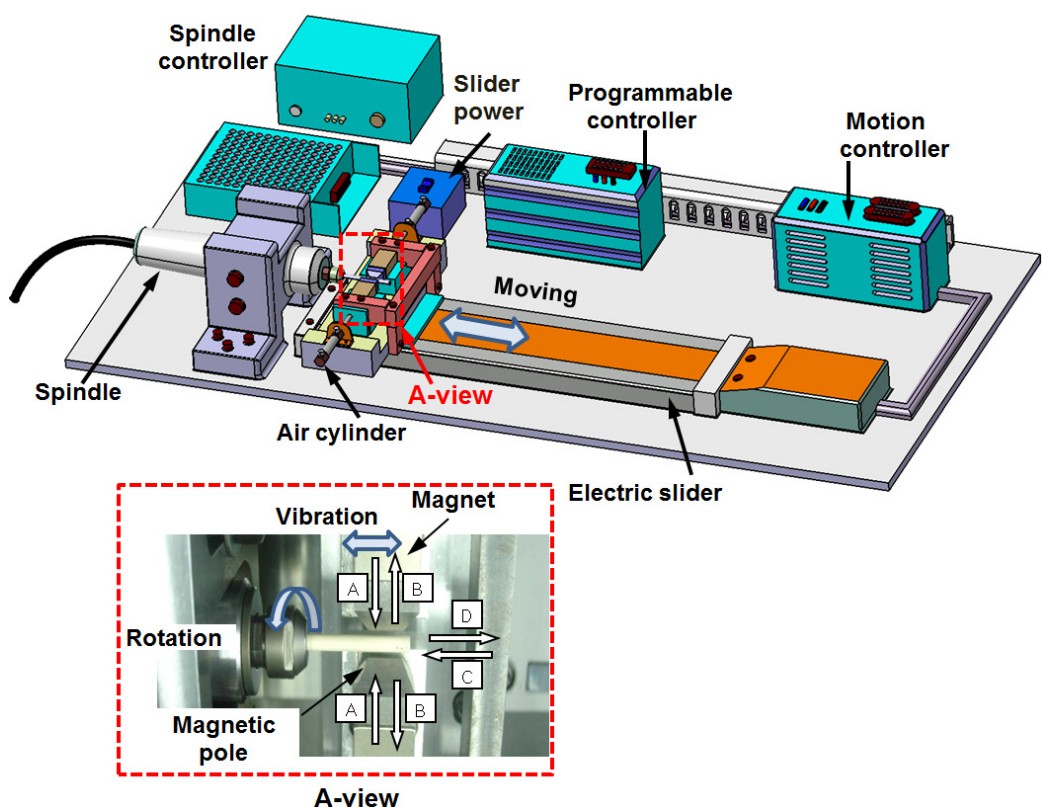

**Figure 2.** High-speed magnetic abrasive finishing system configuration for micro-machining of fine ceramic bar, A-view: Forward/backward directions of a pole in an attempt for the processing.

**Table 1.** Material Characteristics of $Al_2O_3$ fine ceramics.

| Parameter | Conditions |
| --- | --- |
| $Al_2O_3$ Alumina content (%) | 99 |
| Bulk density (g/cm$^3$) | 3.8 |
| Vickers hardness (HV) | 15.2 (load 500 g) |
| Flexural strength (MPa) | 310 |
| Compressive strength (MPa) | 2160 |
| Young's modulus of elasticity (GPa) | 360 |
| Poisson's ratio | 0.23 |
| Thermal conductivity (W/m·K), | 29 (at 20 °C) |

**Table 2.** Experimental conditions.

| Parameter | Conditions |
| --- | --- |
| Workpiece revolution | 30,000 rpm |
| Workpiece | $Al_2O_3$ fine ceramic round bar, $\phi 3 \times 60$ mm <br> Number of workpieces (n = 8) |
| Magnet | Nd-Fe-B permanent magnet: 20 mm $\times$ 10 mm $\times$ 12 mm |
| Magnetic pole <br> Material SS400 | 10   5   20   1   5 5 |
| Magnetic flux density | 0.52 T |
| Vibration of magnetic poles | Amplitude: 2 mm, Frequency(f): 0, 12 Hz |
| Mixed-type magnetic abrasive | Iron particles: 800 mg (mean diameter: 75 μm) <br> Diamond paste weight (w): 50, 200, 400 mg <br> Diamond paste grain size (d.p): 1, 3, 9 μm |
| Lubricant | Grinding fluid: 0.2 mL |
| Clearance | 1 mm |
| Processing time | 270 s |

## 4. Experimental Results and Discussion

### 4.1. Effect of Processing Diamond Grain Size

Figure 3 shows the variation of surface roughness (*Ra*: arithmetical mean deviation of the profile) against processing time in the machined fine ceramic bar. Diamond pastes were used as the abrasive tools for the experiment range from 1 to 3 and 9 μm in grain sizes with a quantity of 200 mg, respectively. These pastes were mixed with 800 mg of electrolytic iron powder and 0.2 mL of grinding fluid prior to actual application for ultra-precision abrasive machining. The findings that are analyzed from the study are as follows: as hinted in the graph, the surface roughness was speedily improved for primary 60 s as for paste grain sizes of 1 and 3 μm; after the duration, the variation of surface roughness becomes low with similarity in all samples. The case that was achievable for the best surface roughness was improved to 0.01 μm when the grain size was 1 μm within 150 s of processing time. However, the slope of surface roughness did not improve further after 150 s. The reason why the roughness did not improve after 150 s is probably because the unevenness was completely removed from the surface of workpiece in 150 s. In case of the 9 μm grain size, the result was far from expectation in comparison with the above two conditions. This can be explained due to the face that the big grain size of magnetic abrasive, such as 9 μm, was likely to cause the betterment of surface roughness being retarded efficiently. When 9 μm was used, the surface roughness of workpiece was improved from 0.42 μm to 0.28 μm within 270 s of processing time. Figure 4 shows the SEM micro images of the workpieces before and after processing with different abrasive grain sizes. Figure 4a–c show the SEM micro-images of workpiece before and after processing by 1, 9 μm of magnetic abrasive, respectively. There are many high peaks and unevenness on the initial surface of workpiece before processing, as shown in Figure 4a. Figure 4b shows the SEM micro-image of workpiece after processing by 1 μm diamond abrasive grain size. As shown in Figure 4b, the high peaks and unevenness were completely removed from the initial surface of workpiece and the surface condition is much smoother the surface condition presented in Figure 4a,b.

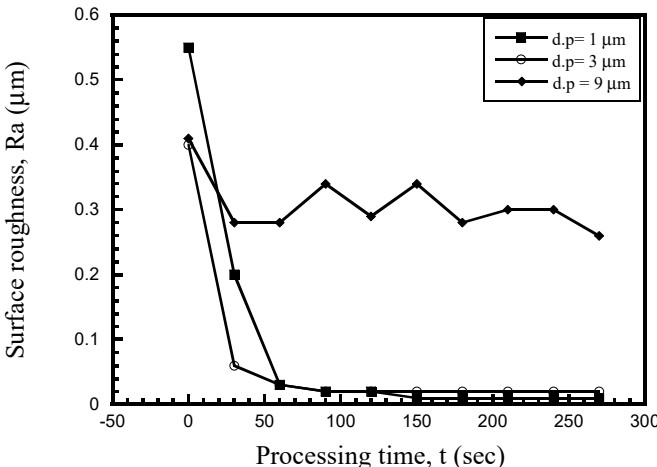

**Figure 3.** Variations of surface roughness (*R*a) against processing time. (spindle revolution: 30,000 rpm, diamond paste weight: 200 mg, frequency of magnetic poles: 12 Hz).

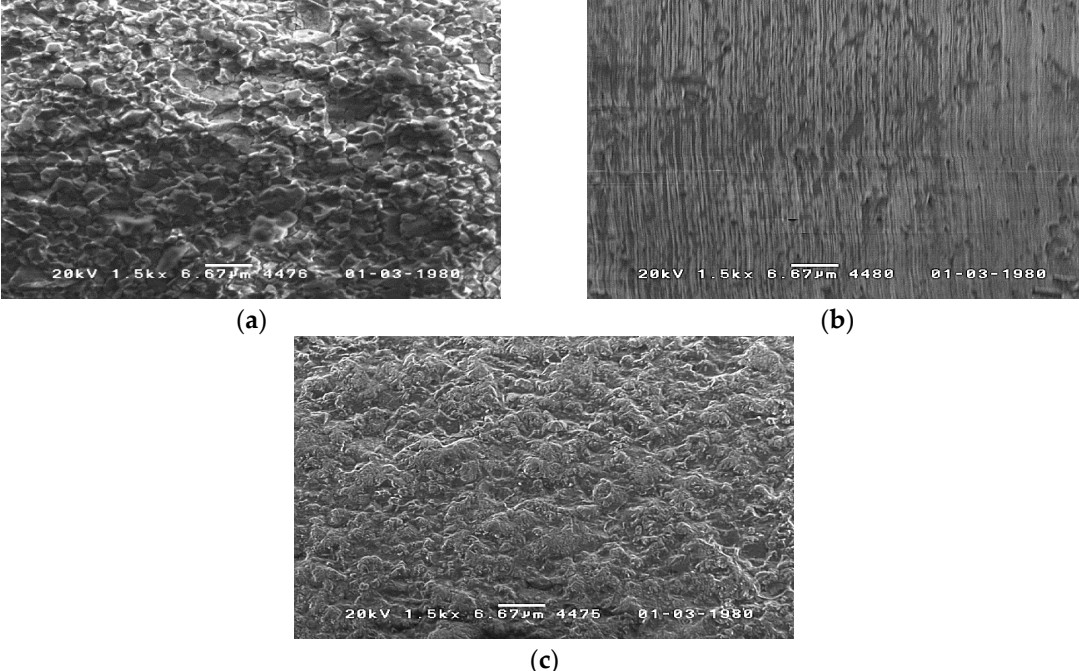

**Figure 4.** Surface conditions of workpieces before and after processing. (**a**) Before abrasive (0 s); (**b**) Diamond paste grain size 1 μm (270 s); and, (**c**) Diamond paste grain size 9 μm (270 s).

Figure 5 shows the improving effect in roundness that is based on least square circle method (LSC), depending on time. Before processing, the original roundness of three workpieces was 2.7 μm, 3.2 μm, and 2.7 μm was processed by 1 μm, 3 μm, and 9 μm, respectively. It was found that the initial sloping for roundness improvement was very big, in that uneven curvature was seemingly removed rapidly by the magnetic abrasive finishing. The cases showing the best achievability in roundness were those of 1 and 3 μm in grain sizes. More specifically, the roundness degree was gain able up to 0.14 μm in elapse of 180 s in case of 1 μm grain size, which showed the same result in 240 s. No other better values could be induced in the case of 1 μm. In the meantime, the roundness degree as for grain size of 9 μm was hardly improved in comparison with the state of 0 s. Among three diamond grain sizes is, consequently, the case of 1 μm, which showed the best achievability in roundness with the least processing time.

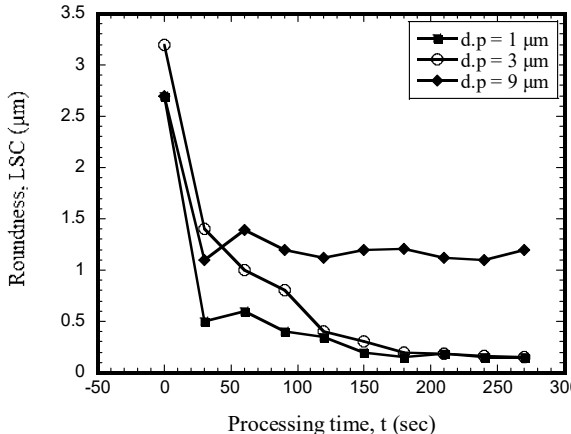

**Figure 5.** Variations of roundness against processing time. (diamond paste weight: 200 mg, frequency of magnetic poles: 12 Hz).

Figure 6 shows the tendency in improving roundness through magnetic abrasive finishing while varying time factors from 0 to 90, 180, and 270 s. Figure 6a indicates the degree of pre-processing roundness, which shows high peaks in pts. A, B, and C, while Figure 6b,c illustrate the improved roundness after processing of 90 and 180 s, respectively. As a whole, peaks of A, B, and C having comparatively bigger unevenness were outstandingly smoothed, owing to improved roundness by abrasive processing. Especially, Figure 6d shows the ultra-precision surface machined after finely removing profile unevenness, which is measured up to the roundness degree of 0.14 μm.

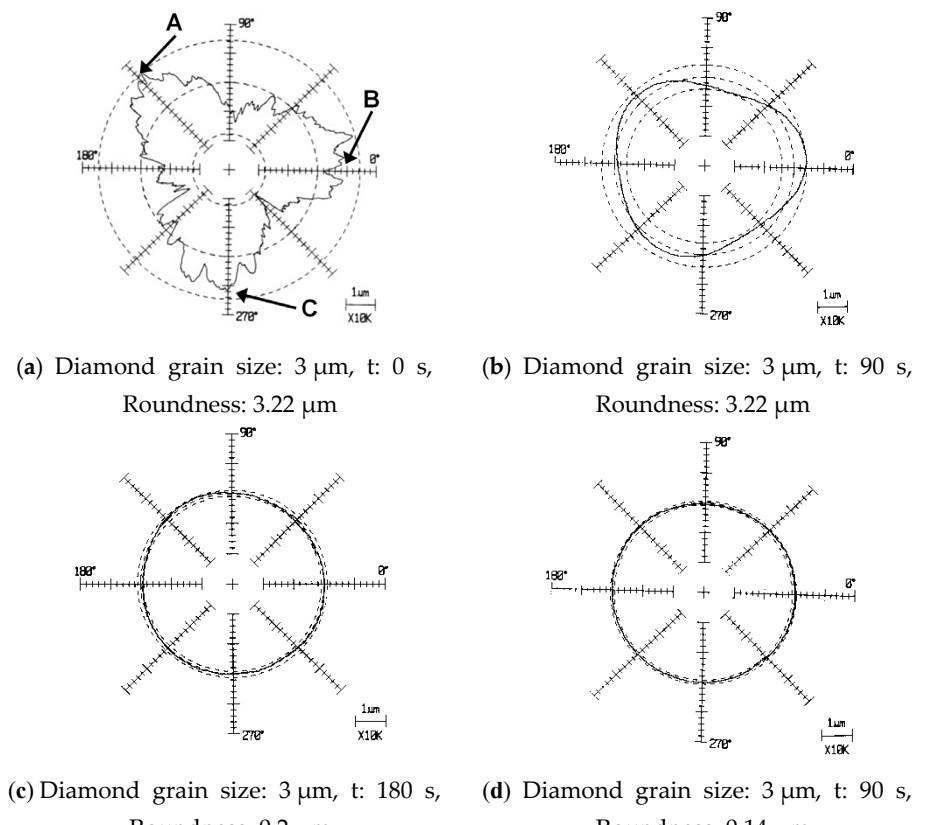

(**a**) Diamond grain size: 3 μm, t: 0 s, Roundness: 3.22 μm

(**b**) Diamond grain size: 3 μm, t: 90 s, Roundness: 3.22 μm

(**c**) Diamond grain size: 3 μm, t: 180 s, Roundness: 0.2 μm

(**d**) Diamond grain size: 3 μm, t: 90 s, Roundness: 0.14 μm

**Figure 6.** Roundness conditions before and after processing as it relates to processing time. (diamond paste weight: 200 mg, frequency of magnetic poles: 12 Hz).

Figure 7 shows the variation in diameters varying when processing was made for each 30 s onto the targeted ceramic bar. In this graph projected for each of three cases as 1, 3, and 9 μm of grain sizes, the diameter changes were divided into two sections: (i) stable region and (ii) unstable region. The vertically dotted line dividing the regions is around the elapse of 90 s. The unstable region exists between 0 and 90 s as shown in the graph where the linear graph plotted with the change in diameters and the data are widely spaced with each other. In particular, the data values are existent far diversely in the case of 9 μm of diamond grain size, which is considered that the uneven profiling factor gives rise to a significant effect on surface finishing for the initial stage. On the other hand, the stable region of diameter change that covers the range of 90 and 270 s is characterized by the linear tendency that diminishes in the diameter change. The diminishing tendency can be expressed with Equations of (1)–(3). In the case of 1 μm grain size, the absolute slope has the value of $0.6095 \times 10^{-4}$ that shows the least tendency in diameter change, being allowable for only 1.8285 μm in each 30 s. This might be caused by the grain size, as a means that its smallest size makes the diameter change lesser comparatively. Furthermore, the diminishing slope as for 3 μm condition shows $1.4849 \times 10^{-4}$ in its absolute value available for processing of 4.4547 μm per 30 s. The slope in case of 9 μm has the value of $3.4009 \times 10^{-4}$ that is the biggest slope. However, this case is found to be unacceptable for ultra-precision abrasive finishing, because its larger grain sizing results in coarse surfacing that leads to not-so-good improvement as per the roundness factor.

$$d_1 = 2.9078 - 6.095 \times 10^{-5} \, t \tag{1}$$

$$d_3 = 2.9013 - 14.849 \times 10^{-5} \, t \tag{2}$$

$$d_9 = 2.8965 - 34.009 \times 10^{-5} \, t \tag{3}$$

where, d: diameter in fine ceramic bar (1, 3, and 9: grain size of diamond), t: processing time (s).

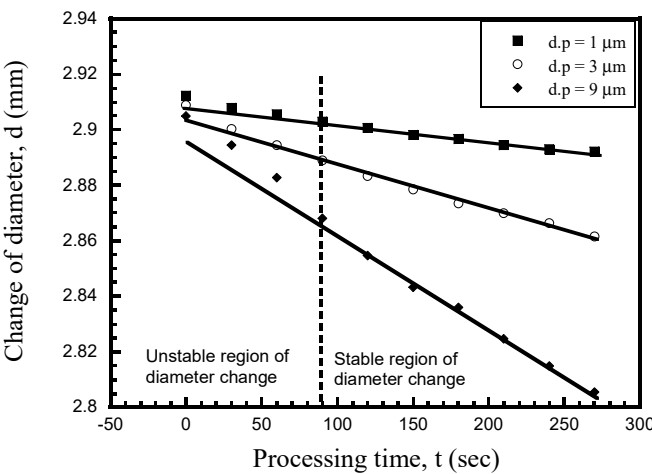

**Figure 7.** Variations of diameter after processing surface diameter against processing time.

Simply speaking, the above graph argues that fine ceramics typically deemed as one of hard-to-grinding materials is sufficiently applicable for magnetic abrasive polishing in order to achieve ultra-precision finishing by modulating diameters linearly in micro dimension within the stable region of diameter change.

### 4.2. Effect of Vibration Frequency

Figure 8 shows the effect of vibration frequency on the change in surface roughness *R*a against processing time. In order to investigate the effect of vibration frequency on change in surface roughness, 0 and 12-Hz of vibration frequency are applied to magnetic poles while using an electric slider. It was

found that the tendency to improve surface roughness for the initial 30 s be accelerated achieved for both two cases. Surface roughness in the sample that had 0 Hz of vibration frequency after 30 s showed no further improvement tendency, being even worse in the duration after 120 s. This can be explained that, without vibration frequency and with only rotational motion of workpiece at 30,000 rpm, the circumferential grooves could be formed on the finished surface finish. In the case of 12 Hz, the surface roughness was improved vividly other than the case of 0 Hz, which was strengthened even up to 270 s. The evidences as above demonstrated that vibration to the longitudinal direction of samples to be polished must be also put into action, even under high speed revolutions of 30,000 rpm for the purpose of obtaining better advanced surface conditions through ultra-precision abrasive finishing.

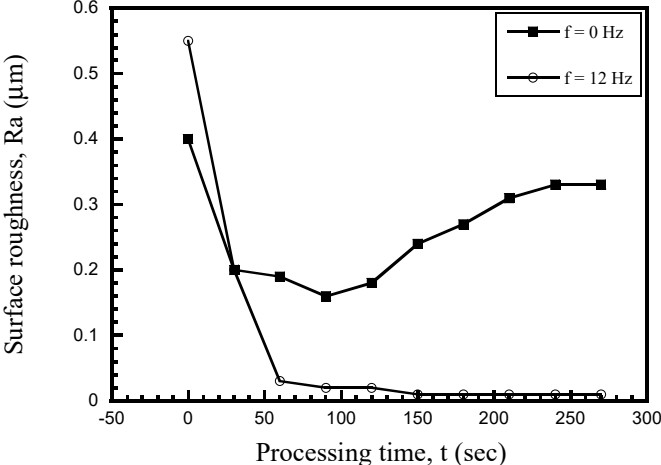

**Figure 8.** Variations of surface roughness (*R*a) against processing time. (spindle revolution speed: 30,000 rpm, diamond paste weight: 200 mg, diamond paste grain size: 1 μm).

Figure 9 presents the variation of diameter depending on processing time, which reveals that in almost all cases, including Hz variation, the diminishing tendency in diameters within the stable region show linear plotting. The variations in cases of 12 Hz and 0 Hz can be expressed by Equations (1) and (4), respectively. It is identifiable that the diminishing slopes as for the case of 12 Hz is bigger than that as for 0 Hz case.

$$d_0 = 2.9066 - 3.714 \times 10^{-5} \, t \tag{4}$$

where, $d_0$: diameter of fine ceramic round bar, f: Vibration frequency of magnetic poles.

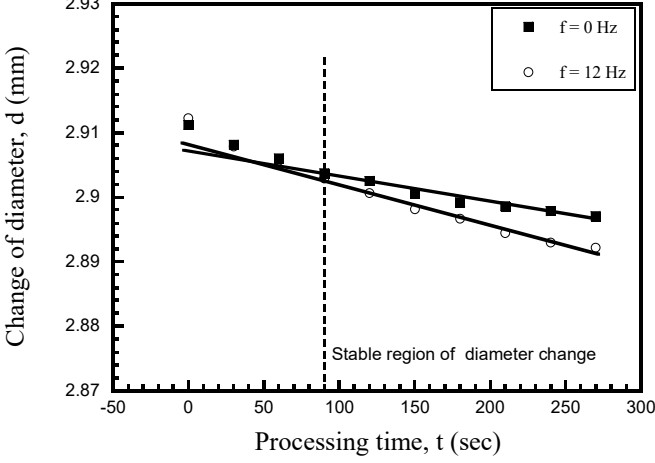

**Figure 9.** Variations of diameter against processing time.

Figure 10 shows the tendency in variations of removed weight against processing time, the values of which are detectably significant in both cases of 0 and 12 Hz for the initial duration of processing. In case of the 12 Hz, the amount of material removed weight was generally largely than 12 Hz of vibration frequency. This can be confirmed that the case of 12 Hz-vibration was applied; the friction between magnetic abrasive particles and surface of workpiece was decreases. Thus, the decrease of friction creates more opportunities for moving more material removed when compared to 0 Hz of magnetic pole vibration frequency.

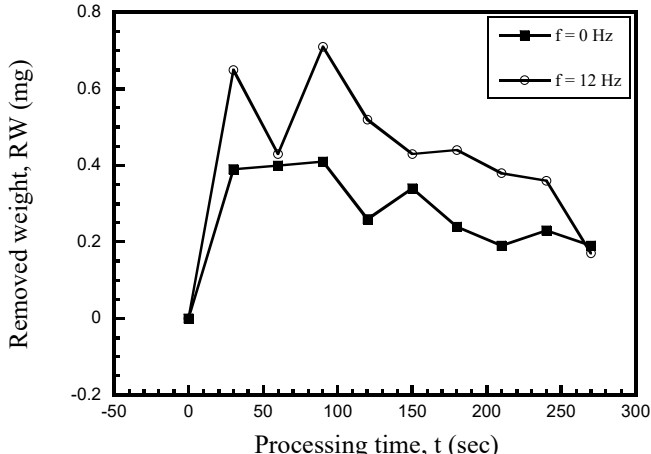

**Figure 10.** Variations of removed weight against processing time.

### 4.3. Effect of the Weight of Diamond Paste

Figure 11 shows the variations of surface roughness varying from quantities of diamond paste. Each of 50 mg, 200 mg, and 400 mg quantities of diamond paste was resultantly gainable for surface roughness in ultra-precision processed surfaces up to the value of 0.01 μm around processing times of 150, 90, and 120 s, respectively. However, when the processing time exceeds the threshold of 90 s, surface roughness seemed to have similar value as a whole. This can be concluded that each amount of diamond paste (50, 200, and 400 mg) have a significant effect on the improvement in surface roughness.

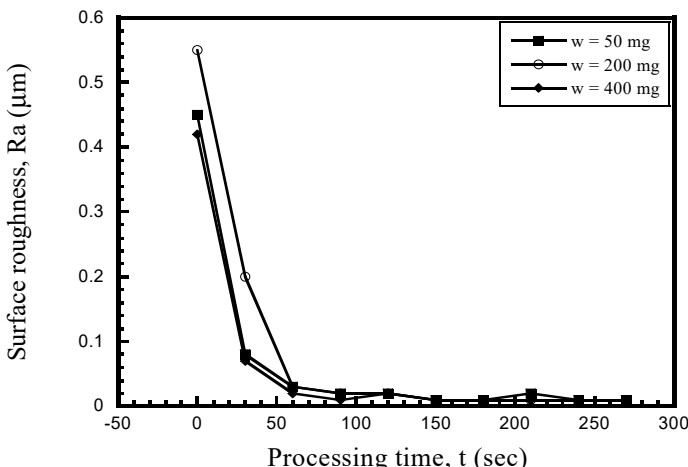

**Figure 11.** Variations of surface roughness (*R*a) against processing time (diamond paste size: 1 μm, frequency of magnetic poles: 12 Hz).

Figure 12 shows the variations of diameter against processing time, which analytically showed that the following (5), (1), and (6) equations with linear characteristics within stable region of diameter

changes be extracted as for cases where weights of diamond grains are 50 mg, 200 mg, and 400 mg. There is little significant difference in their slopes. The average processing capacity in the case of 200 mg weight in diamond paste was 1.829 µm per 30 s; likewise, 1.682 and 1.777 µm as for 50 mg and 400 mg, respectively.

$$d_{50} = 2.9034 - 5.6071 \times 10^{-5} \, t \tag{5}$$

$$d_{400} = 2.8995 - 5.923 \times 10^{-5} \, t \tag{6}$$

where, dw: diameter of fine ceramic round bar (w: weight of diamond paste).

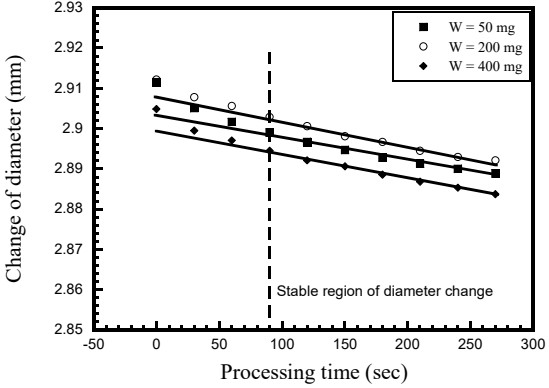

**Figure 12.** Variations of diameter against processing time.

Therefore, the graph affords drawing out the fact that there exists the stable region of diameter change, even by modulating the quantity in diamond paste. Though not being so significant, the slope of diameter change is also affected.

Figure 13 shows the improved variations of roundness against processing time under the conditions that are identical to those in Figure 11. The degree of roundness improved as for 50 mg of diamond paste weight was found to be superior for the segment of 60–150 s in comparison with other cases of 200 mg and 400 mg of diamond pastes, but no further improvement was detected beyond the time segment, while getting up to 0.17 µm in roundness in the case of 270 s. The roundness value of 0.14 µm was measured in time spacing of 270 s as for 400 mg of diamond paste weight. However, the processing time was considerably lessened in the case of 200 mg-diamond paste weight by successfully gaining the same value around 180 s, which highly suggests that the quantity of 200 mg in diamond paste be the most efficient value that is applicable for ultra-precision abrasive polishing.

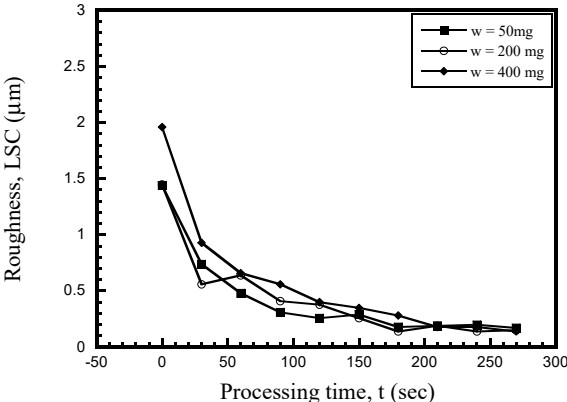

**Figure 13.** Variations of roundness against processing time. (diamond paste size: 1 µm, frequency of magnetic poles: 12 Hz).

Figure 14 shows the variations of the removed weight in targeted workpiece against processing time. It was found that, as for all cases of 50 mg, 200 mg, and 400 mg in weights of diamond paste, the initial volume in weight removed be unstable while the removed weight is not generally affected by the quantities of diamond pastes.

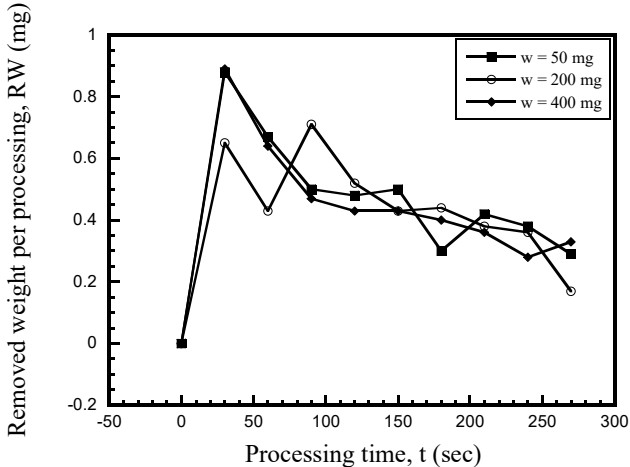

**Figure 14.** Variations of removed weight against processing time. (diamond paste size: 1 μm, frequency of magnetic poles: 12 Hz).

### 4.4. Surface Conditions in Micro-Machining

Figure 15 shows the surface conditions after ultra-precision processing when experimental parameters are preset, as follows: 200 mg of diamond paste weight, 1 μm of grain size, and 12 Hz of vibration frequency. (a) is the non-processed surface condition of the workpiece with very poor reflection degree being hardly legible for the letter X at the bottom side of the ceramic bar. On the contrary, (b) represents the processed surface condition applied by ultra-precision polishing during 270 s that has an excellent reflection degree that is enough to easily read the letter. The value of surface roughness (Ra) was outstandingly improved to 0.01 μm from the original value of 0.55 μm for the short period with more than 50 times, while the degree of roundness was bettered approx. 10 times.

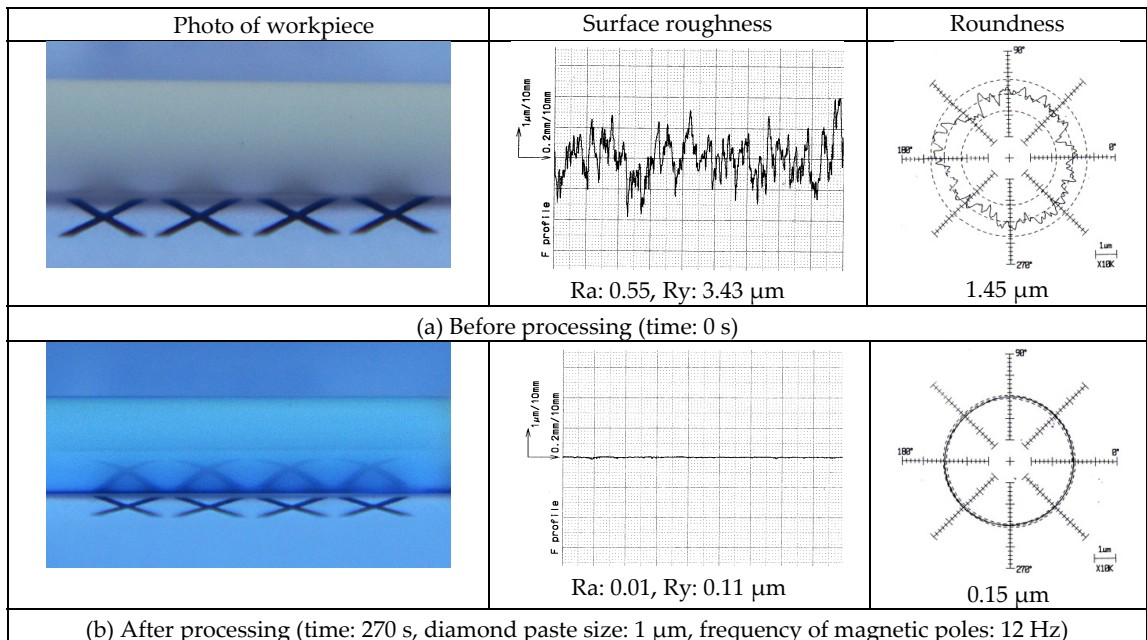

**Figure 15.** Surface conditions after ultra-precision processing.

## 5. Conclusions

In this study, the ultra-precision finishing process was applied through the high-revolution speed of magnetic grinder onto rigid and brittle fine ceramics and the results are drawn, as follows:

1. The magnetic abrasive finishing technique under the condition of high revolution makes it possible to achieve ultra-precision finish in surfaces of fine ceramic material having strong hardness and fragility for grinding, while on the other hand upgrading the roundness for a comparatively short time. Among three sampled conditions of 1, 3, and 9 μm of diamond grain sizes, it was the case of 1 μm that showed the best values as for surface roughness and roundness together with shortest processing time. In this case surface roughness *R*a and roundness (LSC) were measured with the values of 0.01 μm and 0.14 μm, respectively.

2. The tendency changing in diameter could be descriptive by dividing into two segments—stable region and unstable region for diameter change—the former of which allows for realizing ultra-precision processing by controlling the diameter in the workpiece in micro dimension.

3. As far as roundness was concerned, the improving slope detected for the initial stage of processing showed very steep, because non-superficial unevenness in the workpiece prior to processing was rapidly removed due to magnetic abrasive finishing. Additonally, abrading performance was intensively made around the area that had bigger unevenness, accordingly being beneficial for better roundness.

4. In the stable region of diameter change, the case of 12 Hz in vibration frequency showed higher variation magnitude in diameter for the whole range rather than that in case of 0 Hz. It must be noted that the quantity of diamond paste had minimal significant effects on surface roughness, improved roundness, and processed diameter slope in a workpiece.

**Author Contributions:** Design experiment and writing paper Conceptualization, J.S.; methodology, and performed experiments, T.S. and S.D.M.; and investigation and editing, M.S. and S.D.M. All authors have read and agreed to the published version of the manuscript.

**Funding:** This research was funded by NATIONAL RESEARCH FOUNDATION (NRF) of Korea in 2019, (Research Project No. 2019R1F1A1061819). Also, "This research was supported by research funds for newly appointed professors of Jeonbuk National University in 2018".

**Conflicts of Interest:** The authors have no conflicts of interest to declare.

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
