# Peer review of "Micro-Machining Characteristics in High-Speed Magnetic Abrasive Finishing for Fine Ceramic Bar"

_metals, doi:10.3390/met10040464_

Round 1

Reviewer 1 Report

Interesting paper dealing with Micro-machining of the ceramic bar using Magnetic Abrasive Finishing process. Major review comments on the manuscript are given below.

1) The English language needs extensive revision in order to follow the content of the manuscript 

2) The rationale and novelty of the research should be emphasised properly and more clearly 

3) Scientific discussion of the results is absolutely missing. Not a single scientific reference was used to discuss the observed results.

4) Statical data of the results are also missing. For example, how many roughness measurements were performed? No error bars were provided on the graphs.

 All the above comments need to be considered and should be implemented for the betterment of the manuscript. 

Author Response

Dear Editor and Reviewers,

We would like to thank the editor and reviewers for careful and thorough reading of this manuscript and for the thoughtful comments and constructive suggestions, which help to improve the quality of this manuscript.

Reviewer 2 Report

Interesting article. For the future, I propose to extend the analysis of research by surface morphology from which information on its functionality can be obtained.

Author Response

(The authors gave the same response as above.)

Reviewer 3 Report

The manuscript could be accepted for publication after minor revision (corrections to minor methodological errors and text editing).

Author Response

(The authors gave the same response as above.)

Reviewer 4 Report

  1. The introduction should improve. There is no analysis of the existing bibliography. It is necessary to highlight the studies of other authors and compare them with that proposed by the author.
  2. It is necessary to review the English of the entire manuscript, there are errors and misspelled phrases such as: .... “A fine ceramic 70 round bar is positioned to magnetic poles”.... line 70;…. “meanwhile, the magnet was Nd-Fe-B permanent magnet”…line 73.
  3. In Figure 3, why 9 µm is so far from the 3 and 1 µm? It is necessary to clarify this on the manuscript.
  4. Why 12 Hz showed better results when is compared to 0Hz?
  5. Bibliography needs improvement

Author Response

(The authors gave the same response as above.)

Round 2

Reviewer 1 Report

Dear Authors,

The revised version is promising. Good explanation and discussion of the results. I would prefer to have more reference to support or discuss your results. But I can understand that this is a novel study and not many publications are available to support/discuss your results.

Reviewer 4 Report

Thanks for the comments and for the answers provided.